# Arylcarboxylation of unactivated alkenes with $CO_2$ via visible-light photoredox catalysis

Wei Zhang[1,2], Zhen Chen[1], Yuan-Xu Jiang[1], Li-Li Liao[3], Wei Wang[1], Jian-Heng Ye[1] ✉ & Da-Gang Yu [1,4] ✉

Photocatalytic carboxylation of alkenes with $CO_2$ is a promising and sustainable strategy to synthesize high value-added carboxylic acids. However, it is challenging and rarely investigated for unactivated alkenes due to their low reactivities. Herein, we report a visible-light photoredox-catalyzed arylcarboxylation of unactivated alkenes with $CO_2$, delivering a variety of tetrahydronaphthalen-1-ylacetic acids, indan-1-ylacetic acids, indolin-3-ylacetic acids, chroman-4-ylacetic acids and thiochroman-4-ylacetic acids in moderate-to-good yields. This reaction features high chemo- and regio-selectivities, mild reaction conditions (1 atm, room temperature), broad substrate scope, good functional group compatibility, easy scalability and facile derivatization of products. Mechanistic studies indicate that in situ generation of carbon dioxide radical anion and following radical addition to unactivated alkenes might be involved in the process.

Carbon dioxide ($CO_2$), which is inexpensive, non-toxic, and recyclable, has been regarded as an ideal one-carbon feedstock to engage in chemical transformations for the synthesis of high value-added chemicals[1–4]. As carboxylic acids are a privileged functional group in biochemistry and polymer chemistry, it is highly important to develop direct and flexible methods for carboxylation with $CO_2$[5–9]. In recent years, visible-light photocatalytic carboxylation with $CO_2$ has attracted much attention as an efficient, versatile, and sustainable strategy[10–15]. As alkenes are common functional group in organic compounds and bulk chemicals in industry, visible-light photocatalytic carboxylation of alkenes with $CO_2$ is of particular interest[16–29]. Notably, visible-light photoredox-catalyzed difunctionalizing carboxylation of alkenes with $CO_2$ has recently emerged as an important access to valuable carboxylic acids with diverse functionality and high step economy[22–29]. Many groups, including Martin, Wu, Li, Xi, and our group, have reported visible-light photoredox-catalyzed 1,2-difunctionalizing carboxylation of alkenes with $CO_2$ under mild conditions in high chemo- and regio-selectivities (Fig. 1a)[22–29]. However, these methods are mainly

limited to activated alkenes, such as styrenes and acrylates. The photocatalytic 1,2-difunctionalizing carboxylation of unactivated alkenes with $CO_2$ has not been disclosed yet.

As well known, unactivated alkenes are more abundant and easily available in nature and industry than activated alkenes. However, it is challenging for unactivated alkenes to undergo photocatalytic carboxylations with $CO_2$[30–33], arising from high reductive potentials of both starting materials[34–39] and sluggish radical addition onto unactivated alkenes to generate alkyl carbon radicals[40–49], which are less stable than those from activated alkenes. Inspired by our recent work on hydrocarboxylation of unactivated alkenes with $CO_2$[33], we further challenged us whether we could tune the chemoselectivity from C−H to C−C bonds formation based on similar carbon radical intermediates (Fig. 1b). We hypothesized the in situ generation of $CO_2$ radical anion ($CO_2^{•-}$) and following radical addition to unactivated alkenes would result in unstabilized alkyl carbon radicals, which could be further trapped by arenes to generate the C−C bonds. Final rearomatization could give the desired arylcarboxylation products. If successful, it will

[1]Key Laboratory of Green Chemistry & Technology of Ministry of Education, College of Chemistry, Sichuan University, Chengdu 610064, China. [2]West China School of Public Health and West China Fourth Hospital, Sichuan University, Chengdu 610041, China. [3]School of Chemistry and Chemical Engineering, Chongqing University, Chongqing 400030, P. R. China. [4]State Key Laboratory of Elemento-Organic Chemistry, Nankai University, Tianjin 300071, P. R. China. ✉e-mail: jhye@scu.edu.cn; dgyu@scu.edu.cn

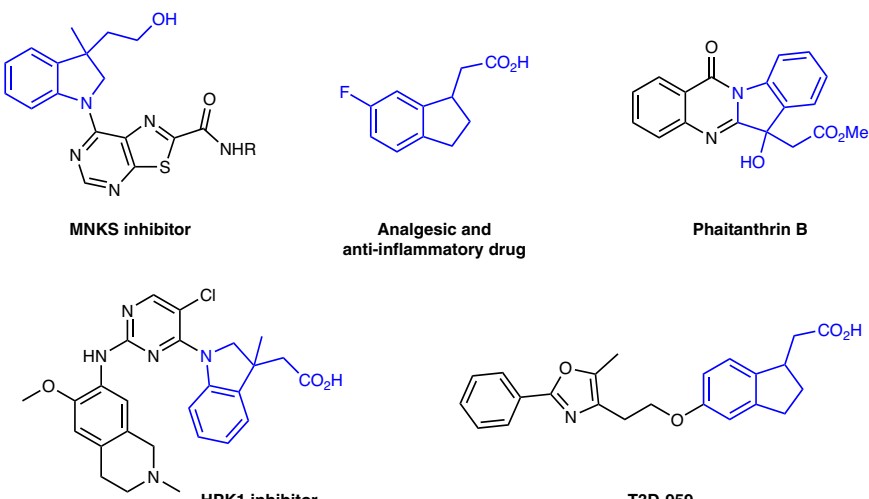

**Fig. 1 | Visible-light photocatalytic 1,2-difunctionalizing carboxylation of alkenes with $CO_2$. a** Visible-light photocatalytic 1,2-difunctionalizing carboxylation of activated alkenes with $CO_2$. **b** Visible-light photocatalytic arylcarboxylation of unactivated alkenes with $CO_2$. PC photocatalyst, EWGs electron-withdrawing groups.

**Fig. 2 | Selected biologically active carboxylic acids and derivatives bearing polycyclic structures.** Examples of biologically active compounds possessing polycyclic acids and derivatives motifs.

realize 1,2-difunctionalizing carboxylation of unactivated alkenes with $CO_2$. Moreover, as it is redox-neutral and atom-economic based on the C−H functionalization, it will also provide a practical and sustainable strategy to access a wide range of polycyclic carboxylic acids, which

are highly important but not easy to obtain via other methods (Fig. 2). Nevertheless, many challenges remain. For example, it is challenging for conversion of $CO_2$ into $CO_2^{·-}$ due to the high reduction potential of $CO_2$ [$E_{1/2}$ ($CO_2/CO_2^{·-}$) = −2.21 V vs SCE][50]. Moreover, the addition of

**Table. 1 | Optimization of reaction conditions[a]**

| Entry | Variations | Yield (%)[b] |
|---|---|---|
| 1 | none | 66 (62) |
| 2 | w/o Ir-1 | n.d. |
| 3 | w/o T1 | n.d. |
| 4 | w/o $Cs_2CO_3$ | n.d. |
| 5 | w/o light | n.d. |
| 6 | $N_2$ instead of $CO_2$ | n.d. |
| 7 | T2 instead of T1 | 62 |
| 8 | $PhMe_2SiH$ as an additive | 86 (83) |
| 9[c] | Ir-2 instead of Ir-1 | 60 |
| 10[c] | 4CzIPN instead of Ir-1 | n.d. |
| 11[c] | DMF instead of DMSO | 55 |
| 12[c] | $^tBuSH$ instead of T1 | 74 |
| 13[c] | $K_2CO_3$ instead of $Cs_2CO_3$ | 68 |
| 14[c] | PMHS instead of $PhMe_2SiH$ | 82 |

*n.d.* not detected, *DMSO* dimethyl sulfoxide, *DMF N,N*-dimethylformamide, *ppy* 2-phenylpyridine, *dtbbpy* 4,4'-di-*tert*-butyl-2,2'-bipyridine, *4CzIPN* 2,4,5,6-tetra(carbazol-9-yl)isophthalonitrile, *PMHS* poly(methylhydrosiloxane).
[a]Reaction conditions: **1a** (0.2 mmol, 1.0 equiv), **Ir-1** (1 mol%), **T1** (20 mol%), $Cs_2CO_3$ (3.0 equiv.), DMSO (2 mL), irradiation by 30 W blue LEDs at room temperature (rt) under $CO_2$ (1 atm) for 24 h.
[b]Yield determined by $^1H$ NMR using 1,3,5-trimethoxybenzene as an internal standard. Isolated yields in parentheses.
[c]$PhMe_2SiH$ (1.0 equiv.) was used.

nucleophilic $CO_2^{\cdot-}$ to electron-rich unactivated alkenes is a polarity-mismatched process[51]. In addition, hydrocarboxylation, arylthiolation, and other competitive side reactions would also hamper the desired difunctionalizing carboxylation.

Herein, we report our success in realizing the visible-light photoredox-catalyzed arylcarboxylation of unactivated alkenes with $CO_2$ (Fig. 1b). A variety of tetrahydronaphthalen-1-ylacetic acids, indan-1-ylacetic acids, indolin-3-ylacetic acids, chroman-4-ylacetic acids and thiochroman-4-ylacetic acids are generated in high selectivities and moderate-to-good yields.

## Results
### Screening of reaction conditions
As carboxylic acids with polycyclic structures are widely found in natural products, drugs and bioactive compounds (Fig. 2)[52–56], we initiated our project with **1a** as standard substrate to generate tetrahydronaphthalen-1-ylacetic acid **2a** as the desired product (Table 1). In the presence of *fac*-Ir(ppy)$_3$ (**Ir-1**) as photocatalyst, 4-*tert*-butylthiophenol (**T1**) as hydrogen atom transfer (HAT) catalyst and $Cs_2CO_3$ as base (Please see the Supplementary Tables 1–5 in Supplementary Information (SI) for more details), the desired

arylcarboxylation product **2a** was obtained in 66% yield with high selectivity (Entry 1). Control experiments revealed that photocatalyst, thiol catalyst, $Cs_2CO_3$, visible light, and $CO_2$ all played essential roles in the reaction (Entries 2–6). The use of *p*-$^tBuC_6H_4SK$ (**T2**) instead of *p*-$^tBuC_6H_4SH$ (**T1**) provided **2a** in comparable yield (Entry 7). To our delight, $PhMe_2SiH$ turned to be a good additive that enhanced the yield of **2a** to 86%, probably owing to the promotion of the $CO_2^{\cdot-}$ generation in the reaction (Entry 8)[57]. A variety of reaction conditions with other photocatalysts, solvents, HAT catalysts, bases, and silanes were also tested to give lower conversions and yields (Entries 9–14).

### Substrate scope
Having established the optimized reaction conditions, we investigated the substrate scope (Fig. 3). A wide variety of electron-donating groups (EDGs) and EWGs were tolerant at the *para*-positions of the arene moiety, providing the desired products **2a**–**2n** in moderate-to-good yields. Substrates containing various functional groups, such as trifluoromethoxyl group (**2d**), fluoro (**2g**), amines (**2i**–**2k**), thioether (**2l**) and amide (**2m**), were smoothly converted to the corresponding products, thus allowing for downstream transformations. The efficiency of this protocol was not hampered by the *ortho* substituents on

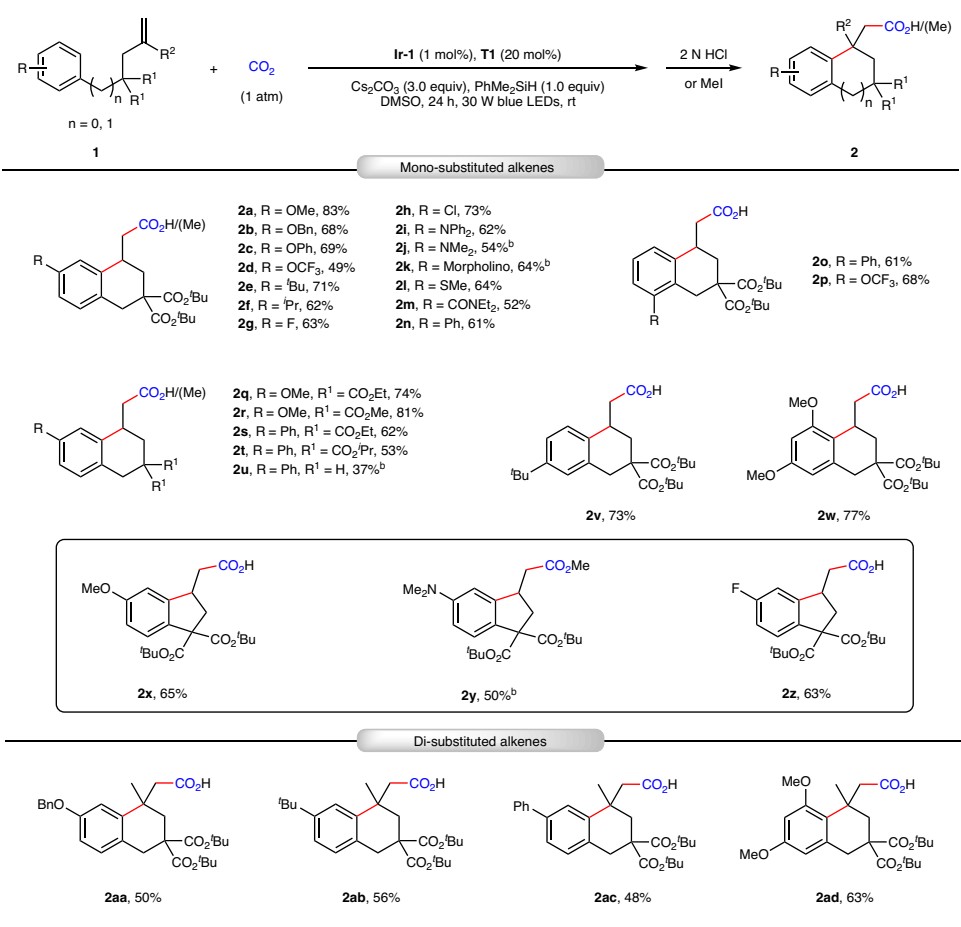

**Fig. 3 | Arylcarboxylation of unactivated alkenes with $CO_2$ to construct tetrahydronaphthalen-1-ylacetic acid and indan-1-ylacetic acid derivatives.** [a]Standard reaction conditions (Table 1, Entry 8) with yields of isolated carboxylic acids or methyl esters. [b]Esterification by MeI (0.4 mmol, 2.0 equiv.), 65 °C, 3 h.

the phenyl ring, giving the corresponding arylcarboxylation products **2o**–**2p** in moderate-to-good yields. Substrates with different substituents on the aliphatic chain were also suitable for such a transformation, furnishing products **2q**–**2t** in 53–81% yields. When no ester group was present in the substrate, the carboxylative cyclization product **2u** could also be obtained. To our delight, substrate **1v** with *tert*-butyl group at the *meta*-position of the phenyl ring was tested in this reaction to give product **2v** in 73% yield and sole regioselectivity owing to the steric hindrance effect. The substrate **1w** bearing di-methoxyl groups also underwent the reaction smoothly to afford the arylcarboxylation product **2w** in 77% yield. We were delighted to find that 5-exo cyclization process could also occur under such conditions, giving the indan-1-ylacetic acids **2x**–**2z** in moderate-to-good yields. We next turned our attention to 1,1-disubstituted unactivated alkenes as $CO_2$ coupling partners, which have rarely been used for photocatalytic cyclization reactions[58]. To our delight, this system also accomplished the 6-exo cyclizations to furnish carbocycles **2aa**-**2ad** containing the quaternary carbon centers in 48–63% yields.

As indoline derivatives are privileged structural motifs found in alkaloids[59] and clinical drugs[60], seeking an efficient and simple approach for the construction of indolines is of continuous interest. Encouraged by the above results, we further turned our attention to selective carboxylation of *N*-protected allylanilines **3** with $CO_2$ to afford indolin-3-ylacetic acid derivatives **4** (Fig. 4). Mono-substituents on the aromatic ring had a negligible impact on these reactions, as the corresponding indoline derivatives **4a**–**4g** were obtained in satisfactory yields. Further investigations of the substrate scope showed that di- or tri-substituted *N*-protected allylanilines also delivered the

corresponding indolin-3-ylacetic acid derivatives **4 h** and **4i** in synthetically useful yields.

Inspired by above results, we wondered whether other kinds of valuable polycyclic carboxylic acids could be formed using this strategy. As chromanes and thiochromanes are widely distributed in nature and display a broad range of biological and pharmaceutical activities[61–63], we further tested phenol- and thiophenol-derived alkenes **5** under standard reaction conditions. Fortunately, these substrates were also reactive to furnish the desired chroman-4-ylacetic acid and thiochroman-4-ylacetic acid derivatives **6a**–**6d** in 21–65% yields (Fig. 5).

## Synthetic applications

In order to demonstrate the utility of this method, a gram-scale reaction and product derivatizations were performed (Fig. 6). The product **2a** was obtained in 84% yield and gram scale, demonstrating the facile scalability of this reaction (Fig. 6a). Then, we carried out the derivatization of **2a** to illustrate potential synthetic applications (Fig. 6b). Selective reduction of product **2a** by using NaBH₄ produced the alcohol **7** in 92% yield[64]. Condensation between **2a** and methyl glycinate hydro-chloride gave cyclic amide **8** in an excellent yield[65]. A practical decarboxylation of primary carboxylic acid **2a** via synergistic photoredox and HAT catalysis was achieved in excellent yield[66]. And **2a** could also participate in decarboxylative trifluoromethylation to give compound **10** in moderate yield[67]. Notably, compound **2a** was easily transformed to the redox-active ester **11**[68], which underwent C−P and C−S bonds formation through decarboxylative phosphination[69] and arylthiolation[70], respectively.

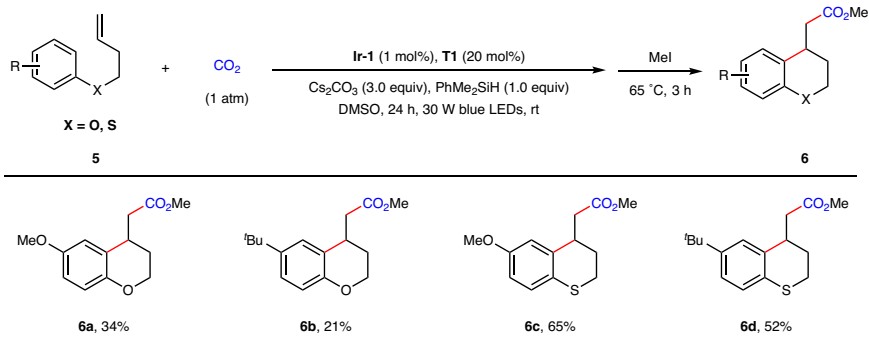

**Fig. 4 | Arylcarboxylation of unactivated alkenes with $CO_2$ to construct indolin-3-ylacetic acid derivatives.** [a]Standard reaction conditions (Table 1, Entry 8) with yields of isolated methyl esters.

**Fig. 5 | Arylcarboxylation of unactivated alkenes with $CO_2$ to construct chroman-4-ylacetic acid and thiochromane-4-ylacetic acid derivatives.** [a]Standard reaction conditions (Table 1, Entry 8) with yields of isolated methyl esters.

### Mechanistic investigations

To gain more insight into this reaction, a series of control experiments were conducted (Fig. 7). When the reaction was performed in the presence of various radical scavengers, such as 2,2,6,6-tetramethyl-1-piperidiny-1-oxy (TEMPO) or diphenyldiselenide (PhSeSePh), the formation of product **2a** was completely inhibited with almost full recovery of **1a**, indicating that radical process might be involved (Fig. 7a). As the formation of reduction product **1a′** was not observed under nitrogen atmosphere, we believed that unactivated alkenes could not be reduced in the reaction (Fig. 7b). The results of detecting of formate ($HCO_2^-$) in the presence or absence of unactivated alkenes indicated that $CO_2^{\bullet-}$ could be generated from single electron reduction of $CO_2$ in the reaction (Fig. 7c). Moreover, Stern-Volmer fluorescence quenching experiments showed that the excited state of the photocatalyst was quenched by the thiolate rather than unactivated alkenes (Fig. 7d).

Based on the control experiments and previous studies[71–73], a possible mechanism for the overall transformation of **1a** is proposed (Fig. 8). The irradiation of photocatalyst *fac*-Ir$^{III}$(ppy)₃ generates excited *fac*-*Ir$^{III}$(ppy)₃ ($E_{1/2}^{*III/II}$ = +0.31 V vs SCE), which can be reductively quenched by a catalytic thiolate to furnish *fac*-Ir$^{II}$(ppy)₃ and a thiyl radical.

Then, the Ir$^{II}$ species ($E_{1/2}^{III/II}$ = −2.19 V vs SCE)[72] may engage in reducing $CO_2$ [$E_{1/2}$ ($CO_2/CO_2^{\bullet-}$) = −2.21 V vs SCE][50] via SET event to deliver $CO_2^{\bullet-}$ along with regeneration of *fac*-Ir$^{III}$(ppy)₃ to close the photoredox catalytic cycle. The in situ generated $CO_2^{\bullet-}$ then undergoes radical addition to the C = C double bond of unactivated alkene of **1a** to form an alkyl carbon radical **A**[30,33], which is supposed to be quickly captured via cyclization to form the radical intermediate **B**. Finally, the carboxylate could be obtained via a HAT process of radical intermediate **B** with the thiyl radical, along with regeneration of the thiol catalyst[74]. The protonation during workup would afford the final product **2a**. Meanwhile, the intermediate **B** might also undergo intermolecular HAT to deliver anti-Markovnikov hydrocarboxylation byproduct **C**[33]. In addition, we reason that the silane can serve as an additive to promote the generation of $CO_2^{\bullet-}$ from an alternative pathway (Please see Supplementary Fig. 18 in SI) [57]. At this stage, we could not exclude other alternative pathways (Please see SI for details)[75,76].

### Discussion

In summary, we have developed the visible-light photoredox-catalyzed arylcarboxylation of unactivated alkenes with $CO_2$. This protocol

**Fig. 6 | Synthetic applications. a** Gram-scale reaction. **b** Product derivatizations. Please see SI for experimental details. Gly-OMe·HCl glycine methyl ester hydrochloride. HOBt 1-hydroxybenzotriazole, EDCI 1-ethyl-3-(3-dimethylaminopropyl) carbodiimide, NPhth phthalimidyl, BTMG 2-*tert*-butyl-1,1,3,3-tetramethylguanidine, DMAP 4-dimethylaminopyridine. DCC Dicyclohexylcarbodiimide, PMDTA Pentamethyldiethylenetriamine.

provides an efficient and facile approach to an array of high-valued polycyclic carboxylic acids, such as tetrahydronaphthalen-1-ylacetic acids, indan-1-ylacetic acids, indolin-3-ylacetic acids, chroman-4-ylacetic acids and thiochroman-4-ylacetic acids. This reaction features mild reaction conditions, broad substrate scope, and good functional group compatibility. Moreover, the derivatization of products could afford diverse valuable polycyclic compounds, which are difficult to access via other protocols. Further applications of $CO_2^{\cdot-}$ and difunctionalizing carboxylation of unactivated alkenes are undergoing in our group.

## Methods
### Synthesis of 2a-2z
To an oven-dried Schlenk tube (25 mL) equipped with a magnetic stir bar was added the unactivated alkenes (0.2 mmol, 1.0 equiv. for solid substrates) and *fac*-Ir(ppy)$_3$ (1 mol%). The tube was moved into the glovebox where was added the Cs$_2$CO$_3$ (0.6 mmol, 195.5 mg, 3.0 equiv.). The tube was sealed and removed from the glovebox, then evacuated and back-filled with CO$_2$ atmosphere for three times. liquid alkenes were added under CO$_2$ atmosphere followed by anhydrous DMSO (2 mL), PhMe$_2$SiH (0.2 mmol, 27.3 mg, 31 µL, 1.0 equiv.), 4-*tert*-butylthiophenol (0.04 mol, 6.7 mg, 7.0 µL, 20 mol%), and the tube was

sealed at atmospheric pressure of CO$_2$ (1 atm). The reaction was stirred and irradiated with a 30 W blue LED lamp (1 cm away, with a cooling fan to keep the reaction temperature at 25–30 °C and keeping the reaction region located in the center of LEDs lamp) for 24 h. Upon completion of the reaction, the reaction mixture was diluted with 3 mL ethyl ester (EA) and quenched by 3 mL 2 N HCl. After adding 10 mL of H$_2$O, the mixture was extracted by EA for five times and the combined organic phases were concentrated in vacuo. The residue was purified by silica gel flash column chromatography (Petroleum/EA/AcOH 10/1/ ~ 5/1 ~ /5/ 10.2%) to give the pure desired product.

### Synthesis of 2aa-2ad
To an oven-dried Schlenk tube (25 mL) equipped with a magnetic stir bar was added the unactivated alkenes (0.2 mmol, 1.0 equiv. for solid substrates) and *fac*-Ir(ppy)$_3$ (1 mol%). The tube was moved into the glovebox where was added the Cs$_2$CO$_3$ (0.6 mmol, 195.5 mg, 3.0 equiv.). The tube was sealed and removed from the glovebox, then evacuated and back-filled with CO$_2$ atmosphere for three times. liquid alkenes were added under CO$_2$ atmosphere followed by anhydrous DMSO (2 mL), PhMe$_2$SiH (0.2 mmol, 27.3 mg, 31 µL, 1.0 equiv.), 4-*tert*-butylthiophenol (0.04 mol, 6.7 mg, 7.0 µL, 20 mol%), and the tube was sealed at atmospheric pressure of CO$_2$ (1 atm). The reaction was stirred

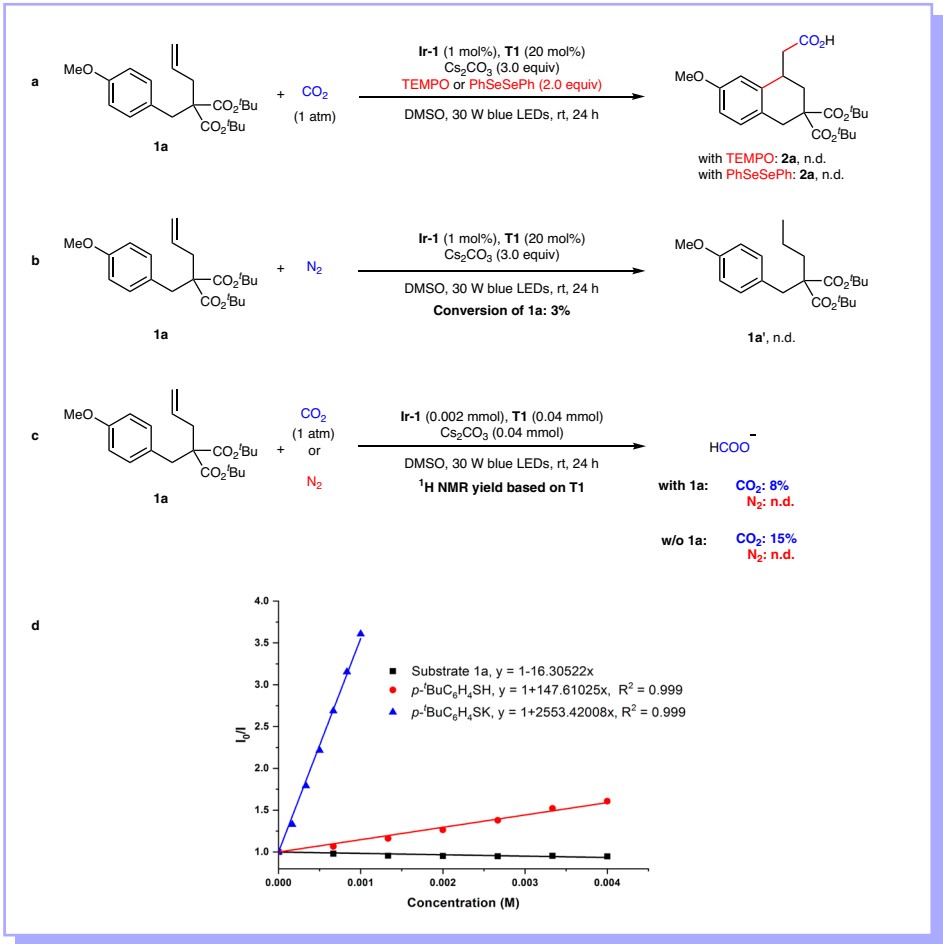

**Fig. 7 | Mechanistic investigations. a** Radical trapping experiments. **b** Reduction of unactivated alkene **1a**. **c** Detection of formate. **d** Stern-Volmer fluorescence quenching experiments.

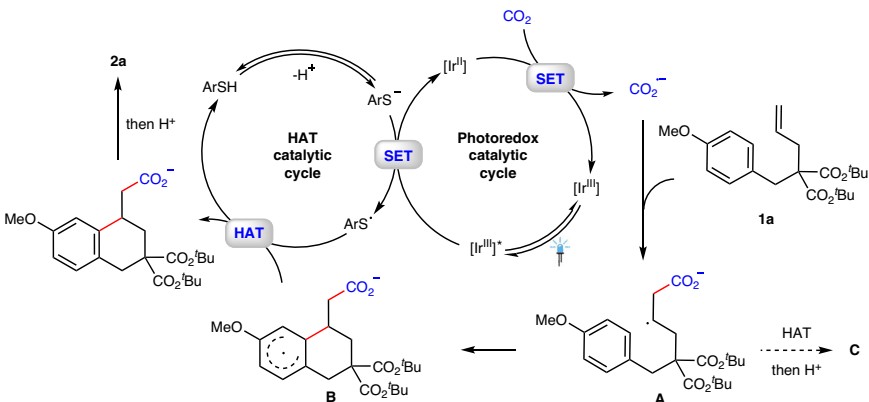

**Fig. 8 | Proposed mechanism.** Proposed catalytic cycle for this synergistic catalyzed arylcarboxylation of unactivated alkenes with $CO_2$.

and irradiated with a 30 W blue LED lamp (1 cm away, with a cooling fan to keep the reaction temperature at 25–30 °C and keeping the reaction region located in the center of LEDs lamp) for 24 h. Upon completion of the reaction, the reaction mixture was diluted with 3 mL EA and quenched by 3 mL 2 N HCl. After adding 10 mL of $H_2O$, the mixture was extracted by EA for five times and the combined organic phases were concentrated *in vacuo*. The residue was purified by silica gel flash column chromatography (Petroleum/EA/AcOH 10/1/ ~ 5/1 ~ /5/10.2%) to give the pure desired product.

### Synthesis of 4a-4i

To an oven-dried Schlenk tube (25 mL) equipped with a magnetic stir bar was added the unactivated alkenes (0.2 mmol, 1.0 equiv. for solid substrates) and *fac*-Ir(ppy)$_3$ (1 mol%). The tube was moved into the glovebox where was added the $Cs_2CO_3$ (0.6 mmol, 195.5 mg, 3.0 equiv.). The tube was sealed and removed from the glovebox, then evacuated and back-filled with $CO_2$ atmosphere for three times. liquid alkenes were added under $CO_2$ atmosphere followed by anhydrous DMSO (2 mL), PhMe$_2$SiH (0.2 mmol, 27.3 mg, 31 μL, 1.0 equiv.), 4-*tert*-

butylthiophenol (0.04 mol, 6.7 mg, 7.0 μL, 20 mol%), and the tube was sealed at atmospheric pressure of $CO_2$ (1 atm). The reaction was stirred and irradiated with a 30 W blue LED lamp (1 cm away, with a cooling fan to keep the reaction temperature at 25–30 °C and keeping the reaction region located in the center of LEDs lamp) for 24 h. Upon completion of the reaction, MeI (0.4 mmol, 25 μL, 2.0 equiv.) was added, the mixture was stirred at 65 °C for 3 h and then cooled to room temperature. The crude reaction mixture was diluted with 3 mL EA. After adding 10 mL of $H_2O$, the mixture was extracted by EA for five times and the combined organic phases were concentrated *in vacuo*. The residue was purified by silica gel flash column chromatography (Petroleum/EA 60/1/ ~ 20/1) to give the pure desired product.

### Synthesis of 6a–6d

To an oven-dried Schlenk tube (25 mL) equipped with a magnetic stir bar was added the unactivated alkenes (0.2 mmol, 1.0 equiv. for solid substrates) and *fac*-Ir(ppy)$_3$ (1 mol%). The tube was moved into the glovebox where was added the $Cs_2CO_3$ (0.6 mmol, 195.5 mg, 3.0 equiv.). The tube was sealed and removed from the glovebox, then evacuated and back-filled with $CO_2$ atmosphere for three times. liquid alkenes were added under $CO_2$ atmosphere followed by anhydrous DMSO (2 mL), PhMe$_2$SiH (0.2 mmol, 27.3 mg, 31 μL, 1.0 equiv.), 4-*tert*-butylthiophenol (0.04 mol, 6.7 mg, 7.0 μL, 20 mol%), and the tube was sealed at atmospheric pressure of $CO_2$ (1 atm). The reaction was stirred and irradiated with a 30 W blue LED lamp (1 cm away, with a cooling fan to keep the reaction temperature at 25–30 °C and keeping the reaction region located in the center of LEDs lamp) for 24 h. Upon completion of the reaction, MeI (0.4 mmol, 25 μL, 2.0 equiv.) was added, the mixture was stirred at 65 °C for 3 h and then cooled to room temperature. The crude reaction mixture was diluted with 3 mL EA. After adding 10 mL of $H_2O$, the mixture was extracted by EA for five times and the combined organic phases were concentrated *in vacuo*. The residue was first purified by silica gel flash column chromatography (Petroleum/EA 150/1/ ~ 60/1) to give the mixture and the yields were determined with $CH_2Br_2$ as an internal standard. The desired arylcarboxylation products were further purified by preparative HPLC.

## Data availability

The authors declare that the data supporting the findings of this study are available within the article and its Supplementary Information files. Extra data are available from the author upon request. The Cartesian coordinates for the calculated structures are available within the Supplementary Data 1.

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

## Acknowledgements

We thank Prof. Yu Lan for helpful discussions. Financial support is provided by the National Natural Science Foundation of China (22225106, for D.G.Y. 22101191, for W.Z. and 22201027 for L.L.L.), Sichuan Science and Technology Program (20CXTD0112, for D.G.Y. and 2021YJ0405 for W.Z.), Fundamental Research Funds from Sichuan University (2020SCUNL102). W.Z. was supported by the China Postdoctoral Science Foundation (2021M692261). We thank Central Government Funds of Guiding Local Scientific and Technological Development for Sichuan Province (2021ZYD0063) and the Fundamental Research Funds for the Central Universities. We also thank Xiaoyan Wang from the Analysis and Testing Center of Sichuan University as well as Jing Li, Qinfang Zhang, and Dongyan Deng from College of Chemistry at Sichuan University for compound testing.

## Author contributions

D.G.Y. and J.H.Y. conceived and designed the study. W.Z., Z.C., Y.X.J., L.L.L., and W.W. performed the experiments, mechanistic studies and wrote the manuscript. All authors contributed to the analysis and interpretation of the data.

## Competing interests

The authors declare the following competing financial interest(s): A Chinese Patent on this work has been applied with the number (202310600327.1). The authors declare no other competing interests.
