## [Peer Review File · Nature Communications]

REVIEWER COMMENTS

Reviewer #1 (Remarks to the Author):

The manuscript describes the first visible-light photoredox-catalyzed arylcarboxylation of unactivated alkenes with CO₂. Some high-valued polycyclic carboxylic acids, such as tetrahydronaphthalen-1-ylacetic acids, indan-1-ylacetic acids, indolin-3-ylacetic acids, chroman-4-ylacetic acids and thiochromane-4-ylacetic acids were prepared in moderate yields. This method shows some advantages including mild reaction conditions, broad substrate scope, and good functional group compatibility. The suggested mechanism seems reasonable. Therefore, I recommends publication in Nat. Commun..

Reviewer #2 (Remarks to the Author):

In this manuscript, Yu and co-workers reported the visible-light photocatalytic arylcarboxylation of unactivated alkenes with CO₂. The reaction has good novelty, mild and green reaction conditions, good functional group tolerance for substrates with electron-donating to electron-withdrawing groups, high selectivity and could afford diverse valuable polycyclic compounds via derivatization. However, the mechanism of this work may need more experiments to support. So this manuscript may need revision before publication with the following suggestions.

[1] As PhMe₂SiH may promote this reaction by participating in the generation of CO₂^{-•}, whether it can be proved experimentally? According to Supplementary Figure 15, HCOOSiR₃ may be the intermediate, could the target product be obtained by directly dropping HCOOSiR₃ instead of PhMe₂SiH to this reaction?

[2] According to the Stern-Volmer fluorescence quenching experiments, the quenching efficiency of p-tBuC₆H₄SK was much higher than p-tBuC₆H₄SH. However, the reaction yield with p-tBuC₆H₄SH was higher than p-tBuC₆H₄SK, how to explain it?

[3] In the proposed mechanism, the CO₂ was reduced by Ir(II) directly, which was uncommon in the previous works. As the reduction potential of Ir(II) (E_{1/2}Ir(II)/Ir = -2.19 V vs SCE) is higher than CO₂ [E_{1/2}(CO₂/CO₂^{-•}) = -2.21 V vs SCE], how to prove the CO₂ was reduced by Ir(II) via usual SET? Other approaches such as conPET may be more reasonable.

[4] The arylation of this reaction is achieved by 6-endo-trig radical addition cyclization. But in many reports there may be a 5-exo-trig radical addition way to synthesis spiro compound (eg. J. Am. Chem. Soc., 2020, 142, 9163–9168.). Is there any 5-exo trig product in this reaction? How the selectivity realized in the radical addition step?

Reviewer #3 (Remarks to the Author):

The authors Yu and Ye et al. have disclosed a newly developed method for arylcarboxylation of unactivated olefins with CO₂ via visible-light photoredox catalysis. A handful of efficient literature precedents are known for photoredox catalysed difunctionalization of activated olefins with in situ generated carbon, phosphorus, silicon, sulfur radical followed by trapping of newly generated carbon radical with CO₂ with good chemo- and regio-selectivities. However, carboxylation of unactivated olefin is challenging due to the high reduction potential of both olefin and CO₂ (>2.2 V vs SCE), which results in sluggish radical addition onto unactivated olefin to generate less stable alkyl carbon compared to those generated from activated olefins. Yu et al. has already reported visible-light photocatalytic di- and hydro-carboxylation of unactivated olefins with CO₂ (Nature Catalysis, 2022, 5, 832–838). This success, along with the insightful designing of the starting material resulted in postulating difunctionalization of unactivated olefin (current work) by further tuning the chemo-selectivity from C–H to C–C bonds formation which enables to overcome competitive side reaction such as hydrocarboxylation, arylthiolation etc. A wide variety of tetrahydronaphthalen-1-ylacetic acids, indan-1-ylacetic acids, indolin-3-ylacetic acids, chroman-4-ylacetic acids and thiochromane-4-ylacetic acids were synthesized in moderate-to-good yields with broad substrate scope and good functional group tolerability. The synthetic utility of this methodology was further demonstrated by diversification of synthesized products to corresponding valuable polycyclic compounds.

Overall, the manuscript is nicely written with detailed mechanistic studies in support of the proposed mechanism, which involves in-situ generation of CO₂ radical anion (CO₂•⁻) followed by CO₂•⁻ radical addition to unactivated olefin resulting an unstabilized alkyl carbon radical, which would undergo addition to arenes to generate the C–C bonds, and thus furnishing difunctionalization of the unactivated olefin. Considering the importance of carboxylation process as well as novelty of this work, the reviewer suggests for the publication of this manuscript in Nature Communications in its present form after minor revision.

The authors are suggested to cite references in support of the following statements:

(i) In page no 5, “To our delight, PhMe₂SiH turned to be a good additive that enhanced the yield of 2a to 86%, probably owing to the promotion of the CO₂•⁻ generation in the reaction (Entry 8).”

(ii) In page no 11.... “Notably, compound 2a was easily transformed to the redox-active ester 10, which underwent C–C, C–P and C–S bonds formation through decarboxylative trifluoromethylation, phosphination, and arylthiolation, respectively.”

Dear Editor, Dear Reviewers,

Thank you very much for handling our manuscript “Arylcarboxylation of unactivated alkenes with CO₂ via visible-light photoredox catalysis” (NCOMMS-23-03312). We have revised the manuscript and SI carefully according to the nice comments and suggestions from editors and the reviewers. Now we submit the revised manuscript along with marks in the final version of the new manuscript. Our responses and revisions are listed point by point according to the corresponding comments as follows:

Reviewer 1

The manuscript describes the first visible-light photoredox-catalyzed arylcarboxylation of unactivated alkenes with CO₂. Some high-valued polycyclic carboxylic acids, such as tetrahydronaphthalen-1-ylacetic acids, indan-1-ylacetic acids, indolin-3-ylacetic acids, chroman-4-ylacetic acids and thiochromane-4-ylacetic acids were prepared in moderate yields. This method shows some advantages including mild reaction conditions, broad substrate scope, and good functional group compatibility. The suggested mechanism seems reasonable. Therefore, I recommend publication in Nat. Commun.

Responses: We thank Reviewer 1 very much for the strong support and nice comments!

Reviewer 2

In this manuscript, Yu and co-workers reported the visible-light photocatalytic arylcarboxylation of unactivated alkenes with CO₂. The reaction has good novelty, mild and green reaction conditions, good functional group tolerance for substrates with electron-donating to electron-withdrawing groups, high selectivity and could afford diverse valuable polycyclic compounds via derivatization. However, the mechanism of this work may need more experiments to support. So this manuscript may need revision before publication with the following suggestions.

Responses: We thank Reviewer 2 very much for the strong support and nice comments! Following these suggestions, we have carefully revised our manuscript. Point-by-point responses are listed below.

[1] As PhMe₂SiH may promote this reaction by participating in the generation of CO₂^{•-}, whether it can be proved experimentally?

Responses: The acceleration effect of PhMe₂SiH could be demonstrated by the detection of HCO₂⁻. A 50% ¹H NMR yield of HCO₂⁻ would be obtained under our standard conditions, whereas only 8% yield of HCO₂⁻ was detected in the absence of PhMe₂SiH. This observation indicated that PhMe₂SiH promoted the reaction by accelerating the generation of HCO₂⁻. For experimental details, please see the SI (Supplementary Figure 3 and Figure 7). Moreover, our previous work has also demonstrated the similar role of hydrosilane in carboxylation reaction with CO₂ (*Chem* **2021**, 7, 3099-3113), and we have included this reference in ref 57 in the

revised manuscript.

According to Supplementary Figure 15, HCOOSiR_3 may be the intermediate, could the target product be obtained by directly dropping HCOOSiR_3 instead of PhMe_2SiH to this reaction?

Responses: We thank Reviewer 2 for the kind suggestion! The control experiments carried out with HCOOSiPhMe_2 instead of PhMe_2SiH under CO_2 and N_2 showed that the product **2a** was produced in 81% and 20% yield, respectively (Figure 1). These results suggest that HCOOSiPhMe_2 might serve as a key intermediate.

Yields determined by ^1H NMR using 1,3,5-trimethoxybenzene as an internal standard and the recoveries of raw materials shown in parentheses.

Figure 1. Determination of the HCOOSiPhMe_2 as a key intermediate

[2] According to the Stern-Volmer fluorescence quenching experiments, the quenching efficiency of $p\text{-}^t\text{BuC}_6\text{H}_4\text{SK}$ was much higher than $p\text{-}^t\text{BuC}_6\text{H}_4\text{SH}$. However, the reaction yield with $p\text{-}^t\text{BuC}_6\text{H}_4\text{SH}$ was higher than $p\text{-}^t\text{BuC}_6\text{H}_4\text{SK}$, how to explain it?

Responses: The reaction yields with $p\text{-}^t\text{BuC}_6\text{H}_4\text{SK}$ and $p\text{-}^t\text{BuC}_6\text{H}_4\text{SH}$ were 66% and 62% respectively, and those yields were comparable. Moreover, the $p\text{-}^t\text{BuC}_6\text{H}_4\text{Cs}$ would be produced by the mixture of $p\text{-}^t\text{BuC}_6\text{H}_4\text{SH}$ and Cs_2CO_3 . Thus, the excited state of the photocatalyst was quenched by the thiolate $p\text{-}^t\text{BuC}_6\text{H}_4\text{Cs}$ other than $p\text{-}^t\text{BuC}_6\text{H}_4\text{SH}$ in our reaction.

[3] In the proposed mechanism, the CO₂ was reduced by Ir(II) directly, which was uncommon in the previous works. As the reduction potential of Ir(II) ($E_{1/2}^{\text{III/II}} = -2.19$ V vs SCE) is higher than CO₂ [$E_{1/2}(\text{CO}_2/\text{CO}_2^-) = -2.21$ V vs SCE], how to prove the CO₂ was reduced by Ir(II) via usual SET? Other approaches such as conPET may be more reasonable.

Responses: We thank Reviewer 2 for this good comment! Calculations were carried out to help understand the reaction profile. The single electron transfer between Ir(II) and CO₂ is 9.7 kcal/mol endergonic, resulting the formation of corresponding CO₂ radical anion and Ir(III) complex. Subsequently, CO₂ radical anion could undergo radical addition to alkene **1a** via transition state **ts1** with an energy barrier of 13.7 kcal/mol, which is exergonic 5.6 kcal/mol and would generate radical **CP2**. The overall activation free energy is 23.4 kcal/mol. Further 6-endo-trig radical addition would occur via **ts2** with an energy barrier of 14.3 kcal/mol to form **CP3** by 3.7 kcal/mol exergonic. Driven by the rearomatization of the intermediate **CP3**, an intermolecular HAT process could take place to deliver the final product by 49.6 kcal/mol exergonic (Figure 2). Overall, the reaction pathway we proposed in Fig 8 in the main text is reasonable.

Figure 2. The DFT calculated free energy profiles for the arylcarboxylation of unactivated alkenes with CO₂ via visible-light photoredox catalysis. The favored pathway is labeled by solid lines. The values given in kcal/mol are the relative free energies calculated by the M06/6-311+G(d,p)//B3LYP/6-31+G(d) method in DMSO solvent.

We also agree with Reviewer 2 that conPET may be a possible pathway and we couldn't exclude this process, which was also proposed in König's work (*Nat. Catal*, 2020, 3, 40-47). In that work, the authors referred that Ir(II) species could be

stabilized by a water/SLES mixture and can further absorb a photon to form Ir(II)*. However, the Ir(II) species decays rapidly in organic solvent like DMSO, which is the solvent used in our reaction. Therefore, the conPET process is difficult to occur in our reaction. Besides, we also performed some mechanistic experiments to identify the ConPET process according to Wu's work (*J. Am. Chem. Soc.* **2021**, *143*, 13266). The details of the results were illustrated as follows:

We first conducted the ^1H NMR investigation (Figure 3) and the characteristic NMR signals of *fac*-Ir(ppy) $_3$ were detected with a mixture of the photocatalyst and *p*- t BuC $_6$ H $_4$ SK in DMSO- d_6 . After exposure of the solutions within the NMR tube to 30W blue LED, the signal was not disappeared. The addition of CO $_2$ showed no effect on the photocatalyst in the absence of light. Overall, these results did not support the conPET process.

Figure 3. ^1H NMR investigation for the possible ConPET process

We next performed the UV-vis spectroscopic study (Figure 4) and no evidence for the formation of new light-absorbing species was observed.

Figure 4. **The emission of *fac*-Ir(ppy)₃ and its mixtures**

At this stage, we have no direct evidence to verify the conPET process. We have updated related statements to the proposed mechanism in the revised manuscript.

[4] The arylation of this reaction is achieved by 6-endo-trig radical addition cyclization. But in many reports there may be a 5-exo-trig radical addition way to synthesis spiro compound (eg. *J. Am. Chem. Soc.*, **2020**, *142*, 9163-9168.). Is there any 5-exo trig product in this reaction? How the selectivity realized in the radical addition step?

Responses: We thank Reviewer 2 for the nice comment! We did not detect the 5-exo trig product under current conditions. To gain more insight into the selectivity issue, we performed the DFT calculations (Figure 5). Once the radical intermediate **CP2** was generated, it could undergo 6-endo-trig radical addition to form intermediate **CP3** or 5-exo-trig radical addition to form intermediate **CP3-1**. Our computational results show that the free energy barrier of 6-endo-trig radical addition via chair transition state **ts2** is 14.3 kcal/mol, and this step is 3.7 kcal/mol exergonic. While the free energy barrier of 5-exo-trig radical addition via transition state **ts2_5-trig-exo** is 15.8 kcal/mol, which is 1.5 kcal/mol higher than that of 5-exo-trig radical addition via transition state **ts2_5-trig-exo**. Besides, this step is 1.3 kcal/mol endergonic. These results clearly suggest that 6-endo-trig radical addition is more thermodynamically and kinetically favorable to occur comparing with 5-exo-trig radical addition. The calculated Mulliken atomic spin densities of transition states **ts2** and **ts2_5-trig-exo**

revealed that the spin density was mainly shared by C1, C3, and C5 of the arene in **ts2**, and by C2 and C4 of the arene in **ts2_5-trig-exo** (Figure 5). The radical in **ts2** is more dispersed than that in **ts2_5-trig-exo**, resulting that **CP3** is more stable than **CP3-1**. Therefore, the stability of the generated radical is responsible for this selectivity.

Figure 5. DFT calculations on the regioselectivity of radical addition cyclization. The values given in kcal/mol are the relative free energies calculated by the M06/6-311+G(d,p)//B3LYP/6-31+G(d) method in DMSO solvent. The favored pathway is labeled by solid lines. The values for the bond length are given in angstroms.

We thank Reviewer 2 very much again for the strong support and nice comments!

Reviewer 3

The authors Yu and Ye et al. have disclosed a newly developed method for arylcarboxylation of unactivated olefins with CO₂ via visible-light photoredox catalysis. A handful of efficient literature precedents are known for photoredox catalysed difunctionalization of activated olefins with *in-situ* generated carbon, phosphorus, silicon, sulfur radical followed by trapping of newly generated carbon radical with CO₂ with good chemo- and regio-selectivities. However, carboxylation of unactivated olefin is challenging due to the high reduction potential of both olefin and CO₂ (>2.2 V vs SCE), which results in sluggish radical addition onto unactivated

olefin to generate less stable alkyl carbon compared to those generated from activated olefins. Yu et al. has already reported visible-light photocatalytic di- and hydro-carboxylation of unactivated olefins with CO₂ (*Nature Catalysis*, **2022**, *5*, 832–838). This success, along with the insightful designing of the starting material resulted in postulating difunctionalization of unactivated olefin (current work) by further tuning the chemo-selectivity from C–H to C–C bonds formation which enables to overcome competitive side reaction such as hydrocarboxylation, arylthiolation etc. A wide variety of tetrahydronaphthalen-1-ylacetic acids, indan-1-ylacetic acids, indolin-3-ylacetic acids, chroman-4-ylacetic acids and thiochromane-4-ylacetic acids were synthesized in moderate-to-good yields with broad substrate scope and good functional group tolerability. The synthetic utility of this methodology was further demonstrated by diversification of synthesized products to corresponding valuable polycyclic compounds.

Overall, the manuscript is nicely written with detailed mechanistic studies in support of the proposed mechanism, which involves in-situ generation of CO₂ radical anion (CO₂^{•-}) followed by CO₂^{•-} radical addition to unactivated olefin resulting an unstabilized alkyl carbon radical, which would undergo addition to arenes to generate the C–C bonds, and thus furnishing difunctionalization of the unactivated olefin. Considering the importance of carboxylation process as well as novelty of this work, the reviewer suggests for the publication of this manuscript in *Nature Communications* in its present form after minor revision.

Responses: We thank Reviewer 3 very much for the strong support and nice comments! Following these suggestions, we have carefully revised our manuscript. Point-by-point responses are listed below.

The authors are suggested to cite references in support of the following statements:

(i) In page no 5, “To our delight, PhMe₂SiH turned to be a good additive that enhanced the yield of 2a to 86%, probably owing to the promotion of the CO₂^{•-} generation in the reaction (Entry 8).”

(ii) In page no 11.... “Notably, compound 2a was easily transformed to the redox-active ester 10, which underwent C–C, C–P and C–S bonds formation through decarboxylative trifluoromethylation, phosphination, and arylthiolation, respectively.”

Responses: We thank Reviewer 3 for the kind suggestion! Following the suggestions, we have cited the corresponding references in the revised manuscript.

We thank Reviewer 3 very much again for the strong support and nice comments!

In addition, we regret that there was an error with the derivatization of the product (Fig 6b) in the manuscript. The starting material of decarboxylative trifluoromethylation is carboxylic acid 2a, rather than redox-active ester 11, we have revised Fig 6b and the related statement in the revised manuscript. In addition, we revised the author information and competing interests in the revised manuscript.

Thank you all very much again for insightful comments and kind suggestions. With the improvements that we have made, we believe that the revised manuscript is now suitable for Nature Communications. Thank you!

Best wishes!

Jian-Heng and Da-Gang

Sichuan University

REVIEWERS' COMMENTS

Reviewer #2 (Remarks to the Author):

The authors carefully revised the article and seriously answered the comments. Detailed mechanism experiments as well as DFT calculations were supplemented. After revision, this manuscript is more accurate. This reviewer recommends it to publish in Nature Communication. Some mechanism experiments and DFT calculations in the answering letter could be added into the supporting information, which may be helpful for the readers.

Dear Editors, Dear Reviewers,

Thank you very much for handling our manuscript “Arylcarboxylation of unactivated alkenes with CO₂ via visible-light photoredox catalysis” (NCOMMS-23-03312). We have revised the manuscript and SI carefully according to the nice comments and suggestions from editors and the reviewers. Now we submit the revised manuscript along with marks in the final version of the new manuscript. Our responses and revisions are listed point by point according to the corresponding comments as follows:

Reviewer 2

The authors carefully revised the article and seriously answered the comments. Detailed mechanism experiments as well as DFT calculations were supplemented. After revision, this manuscript is more accurate. This reviewer recommends it to publish in Nature Communication. Some mechanism experiments and DFT calculations in the answering letter could be added into the supporting information, which may be helpful for the readers.

Responses: We thank Reviewer 2 very much for the strong support and nice comments!

We have added some mechanism experiments (Supplementary Figure 13) and DFT calculations (Supplementary Figure 14) into the revised supplementary information.

In addition, we regret that there was an error in Fig 1 in the main text. The first structure was changed and the related references were modified in the revised manuscript. In addition, we revised the proposed mechanism (Fig 8) and related statement in the revised manuscript.

Thank you all very much again for insightful comments and kind suggestions. With the improvements that we have made, we believe that the revised manuscript is now suitable for Nature Communications. Thank you!

Best wishes!

Jian-Heng and Da-Gang

Sichuan University